# OpenReview forum: "Reasoning Up the Instruction Ladder for Controllable Language Models"
_ICLR.cc/2026/Conference — ICLR 2026 Conference Withdrawn Submission_

### Official Review · Reviewer_xjQg · 2025-10-24

**Soundness:** 2
**Presentation:** 2
**Contribution:** 2
**Rating:** 2
**Confidence:** 4

**Summary:**

This paper addresses instruction hierarchy (IH) in large language models (LLMs)—the ability to follow higher-priority (e.g., system-level) instructions while rejecting conflicting lower-priority (e.g., user) directives. The authors argue that current LLMs fail to respect this hierarchy, especially under adversarial inputs (prompt injection, jailbreaks), because they lack explicit reasoning over instruction sources.

They propose Reasoning for Instruction Hierarchy (IH reasoning), which reframes the problem as a meta-reasoning task: the model first reasons about the relationship between system and user prompts, and then decides which to obey. To enable this, they construct VerIH, a synthetic dataset that extends the RLVR-IFEval dataset with both aligned and conflicting system–user prompt pairs whose outputs have verifiable correctness functions. Using Group Relative Policy Optimization (GRPO) and a small “SysHint” instruction encouraging explicit reasoning, they finetune reasoning-capable models (Qwen3-4B/8B, Phi-4-mini-reasoning) on VerIH.

**Strengths:**

* A simple and effective method: Lightweight dataset + RLVR yields measurable performance gains with only ~7K examples.
* Broad benchmark coverage: Includes both in-domain (IHEval, IFBench) and out-of-domain (safety, jailbreak) tests.

**Weaknesses:**

* Incremental novelty: The paper extends earlier instruction hierarchy and reasoning-for-safety works but doesn’t fundamentally rethink model architecture or training beyond RLVR on synthetic conflicts.
* Synthetic dataset limitations: VerIH conflicts are LLM-generated and may lack realism or linguistic diversity; unclear if models overfit to the structure of these synthetic conflicts.

* Evaluation limitations:
    * Heavy reliance on automated verification or guard scoring; limited human evaluation.
    * Safety generalization gains could arise from exposure to adversarial-style constraints, not genuine reasoning over roles.

**Questions:**

* How realistic are VerIH conflicts compared to real-world multi-role instructions (e.g., multi-turn, multi-agent conversations)?
* Could the improvements on safety benchmarks be explained by simple refusal pattern learning rather than hierarchical reasoning?
* How does the model perform when system and user instructions are both benign but subtly contradictory (e.g., stylistic or prioritization differences)?
* Will VerIH be publicly released with generation scripts to verify reproducibility claims?
* Can the approach extend to three-level hierarchies (system–tool–user), or does performance degrade?
* How does the approach compare to embedding separation methods (Wu et al., 2024b) when evaluated under the same GRPO setup?

---

> ### Author Response · Authors · 2025-11-21
> **Rebuttal**
>
> > How realistic are VerIH conflicts compared to real-world multi-role instructions (e.g., multi-turn, multi-agent conversations)?
>
> Please refer to the “Contributions, Generalization/Significant Gain”. The main contribution is not the complexity of the VerIH dataset. It is our method can provide huge gains with a simple dataset.
>
> VerIH only contains simple conflicts without multi-role / multi-turn instructions. But the performance can generalize to multi-turn cases. There is a subset of multi-turn in IHEval:
>
> | Qwen3-8B | Multi-Turn Aligned | Multi-Turn Conflict |
> |-----------------------|-----------|-----------|
> | w/o CoT    	| 85.19% | 29.79% |
> | w/ CoT     	| 87.58% | 28.63% |
> | w/ CoT + SysHint | 87.90% | 40.63% |
> | +VerIH (Ours)   | **92.25%** | **84.53%** |
>
> > Could the improvements in safety benchmarks be explained by simple refusal pattern learning rather than hierarchical reasoning?
> Please refer to the “Contributions, Generalization”
>
> No. As mentioned in the paper, there is no safety-related data during RLVR training (line 098). The model does not learn the refusal pattern at all. The instruction hierarchy reasoning ability generalizes to the safety domain by itself.
>
> > How does the model perform when system and user instructions are both benign but subtly contradictory (e.g., stylistic or prioritization differences)?
>
> Please refer to the “Contributions, Generalization”. This is our RLVR training setup.
>
> > Will VerIH be publicly released with generation scripts to verify reproducibility claims?
>
> Yes. In the section “REPRODUCIBILITY STATEMENT”, we state that VerIH and data generation scripts are published in the provided anonymous Github link.
>
> > Can the approach extend to three-level hierarchies (system–tool–user), or does performance degrade?
>
> Please refer to “General Response, Multi-level Instruction Hierarchies.”
>
> > How does the approach compare to embedding separation methods (Wu et al., 2024b) when evaluated under the same GRPO setup?
>
> Their method is based on SFT without reasoning. And their model is trained on safety datasets.
>
> Their method is orthogonal to ours. They can complement each other.
>
> But based on the results from “General Response, SFT Results” and “Contributions, Significant Gain”, their method is expected to be easily overfit with a small data size. Their method requires adding extra embeddings to the model, which naturally requires retraining the entire post-training stack. Our method is more lightweight.

---

> ### Author Response · Authors · 2025-11-27
>
> Dear Reviewer xjQg,
>
> Thank you for your effort to review our work. We understand that you might be busy recently, but we hope you could take some time to review our rebuttal. We are happy to clarify any points as needed. If our rebuttal addresses your concerns, we would be grateful if you could reflect that in an updated score.

---

### Official Review · Reviewer_xWG4 · 2025-10-27

**Soundness:** 2
**Presentation:** 2
**Contribution:** 2
**Rating:** 4
**Confidence:** 4

**Summary:**

This paper addresses the challenge of enforcing instruction hierarchies in large language models (LLMs), specifically ensuring that system-level directives reliably override user requests in cases of conflict. The authors present VerIH, a newly constructed dataset of system/user prompt pairs (both aligned and intentionally conflicting), with verifiable constraints to facilitate deterministic evaluation. Models are finetuned using reinforcement learning with variable reward (RLVR) on VerIH to enable explicit meta-reasoning over instruction priorities. Experiments across multiple LLMs show improved compliance with instruction hierarchies and enhanced robustness against adversarial inputs, such as prompt injection and jailbreak attacks, while maintaining general reasoning ability.

**Strengths:**

1. Concrete Reformulation of a Critical Problem: The paper offers a clear and well-motivated reformulation of instruction hierarchy resolution as a reasoning problem, supported by real-world motivating scenarios. This framing addresses a persistent weakness in LLM deployment around controllability and safety.
2. VerIH Dataset with Verifiable Constraints: Construction of the VerIH dataset enables systematic training and evaluation. By creating both aligned and conflicting system-user prompt pairs with verifiable, automatable scoring, the approach yields a genuinely measurable target for RL-based learning.
3. Methodological Rigor: The proposed RLVR finetuning with Chain-of-Thought (CoT) reasoning and explicit "SysHint" instructions is well operationalized and clearly described. The role of various system prompts and ablation variants is made explicit.
4. Empirically Grounded Claims with Extensive Evaluation: The experimental section covers a diverse set of benchmarks, including general instruction following (IFEval, IFBench), instruction hierarchy (IHEval), safety, and general reasoning.
5. Robust Generalization: The paper demonstrates that training on general instruction hierarchy tasks in VerIH out-of-distribution generalizes to safety-critical applications without explicit safety data (Table 2). This supports the claim that explicit reasoning about instruction hierarchies yields a flexible and robust mechanism for model controllability.
6. Analytical Depth and Ablation Analyses: The analysis includes careful ablations isolating the effects of reasoning traces and conflicting prompts, plus an explicit study relating reasoning “coverage” to task performance.

**Weaknesses:**

1. Limited Theoretical Analysis and Justification of Meta-Reasoning Efficacy: While the intuition for meta-reasoning over instruction hierarchies is plausible, the paper lacks a formal or semi-formal analysis or even a taxonomy of potential failure cases for instruction prioritization. For instance, there is no attempt to systematically dissect why explicit reasoning works better than implicit mapping for instruction hierarchy, nor to quantify its limitations (Section 2 and analysis in Section 6 are largely empirical/descriptive).
2. Supervision Signal and Potential RL Pitfalls Underexplored: The reward function Freward used in RLVR training is described in general terms (“verifiable answer constraints”, Section 3), but deeper discussion of its capacity, potential brittleness, or the risk of reward hacking/overfitting is minimal. There is no analysis of reward function coverage, adversarial manipulation resistance, or sensitivity of resulting models to reward definition.
3. Generality Limited by Data Construction Process: The conflicting prompt pairs in VerIH are generated by rewriting user prompts with an LLM. While this approach provides scale, it limits the complexity and naturalness of conflicts. The“conflicting”instructions may be somewhat artificial or simplistic，see the conflict rewrites in Appendix A, where many are mere format or scope differences. It is unclear if the method will hold on truly complex, nuanced, or human-elicited hierarchical instruction conflicts.
4. Result Interpretation and Failure Analysis: While results in Table 1 and Table 2 are generally positive, the paper does not sufficiently analyze failed or ambiguous cases. Which specific types of hierarchical or adversarial conflicts still“slip through”after VerIH RLVR training? The increase in attack success rate for certain settings (noted for Phi-4-mini-reasoning, Table 2) is only superficially discussed.

**Questions:**

1. Can the authors provide deeper theoretical justification (beyond empirical results) for why explicit meta-reasoning on instruction hierarchies achieves more robust compliance than input-response mapping? What are the boundaries of this approach?
2. What specific strategies were used to ensure realistic and challenging “conflicting” instruction pairs in VerIH, beyond formatting/scope variations? Do the authors have examples where existing conflict generation methods fail to produce non-trivial, subtle conflicts?
3. How does the reward function Freward handle ambiguous responses, partial compliance, or compositional system-user conflicts? Could the RL process reward superficial but noncompliant outputs (reward hacking)?
4. Have the authors explored compositional or multi-level hierarchies (more than 2 layers/roles)? How does the approach generalize, and are there examples in VerIH or evaluation where deeper nesting reveals new limitations?
5. In Table 2, Phi-4-mini-reasoning shows higher attack success rates on certain safety benchmarks. Can the authors clarify the cause (overfit, model limitations, or tuning), and suggest remedies or further experiments?

---

> ### Author Response · Authors · 2025-11-21
> **Rebuttal**
>
> > Which specific types of hierarchical or adversarial conflicts still“slip through” after VerIH RLVR training?
>
> Please refer to “Appendix F: Failure cases”.
>
>
> > Can the authors provide deeper theoretical justification (beyond empirical results) for why explicit meta-reasoning on instruction hierarchies achieves more robust compliance than input-response mapping? What are the boundaries of this approach?
>
> We do not think theoretical analysis applies to our paper. But generally, 1) Just like CoT improves LLMs’ performance on math questions, explicit reasoning about instructions helps resolve logic conflicts. 2) During explicit reasoning, the model may generate an integrated task without conflict by combining multi-level instructions before executing the task. This may help with the IH tasks.
>
> > What specific strategies were used to ensure realistic and challenging “conflicting” instruction pairs in VerIH, beyond formatting/scope variations? Do the authors have examples where existing conflict generation methods fail to produce non-trivial, subtle conflicts?
>
> We do not run any checks on the generated dataset. Our paper proved that a simple method with a small noisy dataset can provide a huge improvement. Please refer to “Contributions, Significant Gain”.
>
> And it is resource-intensive to validate the dataset. A careful construction could improve things even more, and we leave that for the future.
>
> > How does the reward function Freward handle ambiguous responses, partial compliance, or compositional system-user conflicts? Could the RL process reward superficial but noncompliant outputs (reward hacking)?
>
> In our design, the reward is given by a verifiable function. So the response can be deterministically verified. The responses must 1) follow the system prompt, and 2) follow the part of the user prompt that does not conflict with the system prompt. Only in this case, a reward will be given. Otherwise, no reward (0 or 1 reward).
>
> Please refer to “Contributions, Generalization”. Our method does not reward hacking since it can generalize to the safety domain, which is not included in the training data.
>
> Reward hacking shows up in the ablation study. There is a case in Table 3, Phi-4-mini-reasoning model, w/o CoT_train setup, on IFBench: “In an exception, for Phi-4-mini-reasoning, the w/o CoTtrain variant improves on IFBench, reaching 45.37%. Closer inspection of model outputs reveals overfitting to prompt-format constraints, causing the model to disregard other instructions and produce meaningless fragments instead of a full sentence (examples in Appendix F).” (line 371)
>
> > Have the authors explored compositional or multi-level hierarchies (more than 2 layers/roles)? How does the approach generalize, and are there examples in VerIH or evaluation where deeper nesting reveals new limitations?
>
> Please refer to “General Response, Multi-level Instruction Hierarchies”.
>
> > In Table 2, Phi-4-mini-reasoning shows higher attack success rates on certain safety benchmarks. Can the authors clarify the cause (overfit, model limitations, or tuning), and suggest remedies or further experiments?
>
> As we mentioned in Section 5 Results line 311, “Phi-4-mini-reasoning is originally optimized primarily for mathematical reasoning, with only a small fraction of its training corpus covering nonmathematical or non-coding reasoning tasks.”
>
> There is few safety-related training corpus used for reasoning training when they released this model. Remedies can be made by further training on safety-related datasets with Phi-4-mini-reasoning.

---

> ### Author Response · Authors · 2025-11-27
>
> Dear Reviewer xWG4,
>
> Thank you for your effort to review our work. We understand that you might be busy recently, but we hope you could take some time to review our rebuttal. We are happy to clarify any points as needed. If our rebuttal addresses your concerns, we would be grateful if you could reflect that in an updated score.

---

### Official Review · Reviewer_i79i · 2025-10-30

**Soundness:** 3
**Presentation:** 3
**Contribution:** 2
**Rating:** 2
**Confidence:** 4

**Summary:**

This paper addresses the critical problem of instruction hierarchy (IH) in Large Language Models (LLMs), where models must reliably prioritize higher-level instructions (e.g., system prompts) over potentially conflicting lower-level instructions (e.g., user prompts). The authors argue that current models fail at this because they treat instructions as a simple input-response mapping problem, leading to vulnerabilities like jailbreaking and prompt injection.


The paper's core contribution is to reframe IH resolution as an explicit meta-reasoning task. Instead of just responding, the model is trained to first "think" about the relationship between the system and user instructions, identify conflicts, and reason about which instruction takes precedence before generating a final answer.

To achieve this, the authors introduce two key components:


* VerIH: A new dataset of 7K instruction-following tasks with verifiable answers. It is built by taking an existing dataset (RLVR-IFEval) and using an LLM (Claude-4-Sonnet) to rewrite half of the user prompts to create explicit conflicts with the system prompts.

* Training Methodology: The authors use lightweight reinforcement learning (specifically RLVR with GRPO) to fine-tune reasoning-enabled models (Qwen3 and Phi-4-mini). The models are trained to generate a chain-of-thought (CoT) reasoning trace within \<think\> tokens before their answer, with a reward signal based on the correctness of the final (verifiable) answer.


The paper claims that this approach significantly improves models' ability to follow instructions and resolve hierarchies, especially in conflicting scenarios. The most significant claim is that this learned reasoning ability generalizes out-of-distribution (OOD) to safety-critical settings. By simply adding safety policies as a high-priority system prompt at inference time, the trained models show enhanced robustness against jailbreak and prompt injection attacks, despite never having seen safety-related data during training.

**Strengths:**

* **Novel and Effective Problem Framing:** The key insight to treat instruction hierarchy as a meta-reasoning task rather than a standard alignment problem is a strong and novel contribution. This moves the field beyond implicit learning and toward explicit, scrutable conflict resolution.


* **High-Quality Dataset (VerIH):** The creation of the VerIH dataset is a valuable contribution to the community. The methodology of generating conflicts from an existing verifiable dataset (RLVR-IFEval) is clever, as it preserves the original verification functions and provides a clear reward signal for the RL process.


* **Strong Empirical Results:** The paper demonstrates clear and significant performance gains on instruction hierarchy benchmarks, particularly on the IHEval-conflict set (e.g., +22.87% for Qwen3-4B). This is achieved while maintaining or even slightly improving performance on general reasoning tasks like MMLU and MATH-500, showing the training is targeted and does not cause degradation.


* **OOD Generalization to Safety:** This is the most compelling result of the paper. The ability to train a model on general, non-safety-related constraint conflicts (e.g., "use 8 highlights" vs. "use no formatting") and have that skill transfer to rejecting harmful adversarial prompts (Table 2)  is a significant finding. It supports the hypothesis that safety is a special case of IH and provides a practical path toward building more controllable and dynamically configurable models (e.g., via GuardRules ).

**Weaknesses:**

* **Scalability of Hierarchy:** The paper simplifies the IH problem to two levels: system and user. While it claims the method is "inherently scalable", this is asserted without proof. Real-world applications involve more complex hierarchies (e.g., developer system prompts, user-level system prompts, tool instructions, user data) that may have more nuanced precedence rules. The experiments do not test this scalability.



* **Diversity of Conflicts:** The VerIH dataset's conflicts are generated entirely by Claude-4-Sonnet following a specific prompt template. This could introduce a lack of diversity or an unknown bias in the types of conflicts generated. It is unclear if the model is learning to resolve conflicts in general or just the style of conflicts produced by the generator model.

* **Reliance on Scaffolding:** The method's success seems tied to specific scaffolding: the <think> tokens and the SysHint prompt that explicitly tells the model to reason about conflicts. It is not entirely clear whether the model has acquired a general meta-reasoning skill or simply learned to follow the SysHint prompt effectively.


* **Justification for RL:** The paper uses RLVR/GRPO, which is inherited from the base dataset. However, it is not clearly justified why this is superior to simpler methods. For example, one could simply perform supervised fine-tuning (SFT) on (CoT, Answer) pairs that are known to receive high rewards from the verification function. This might achieve similar results with less complexity.

**Questions:**

1. You claim the two-level (system vs. user) method is "inherently scalable". Could you elaborate on how you envision this working for a more complex, multi-level hierarchy (e.g., System > Tool > User)? Would this require a more complex SysHint prompt, or a dataset with more complex, multi-level conflicts?



2. The use of RLVR/GRPO  is a core part of the methodology. Have you experimented with a simpler SFT approach, where you only fine-tune on high-reward (CoT, Answer) pairs generated from the VerIH dataset? This would help clarify if the full, complex RL loop is necessary or if SFT on curated "good" reasoning traces is sufficient.


3. The VerIH dataset's conflicts are generated by a single LLM (Claude-4-Sonnet). Have you analyzed the diversity of the generated conflicts? Is there a risk that the model has overfitted to the style of conflict that Claude generates, rather than a general concept of instruction conflict?


4. The w/o CoT_train ablation removes reasoning entirely. What happens in an ablation that keeps CoT generation but removes the specific SysHint prompt? This would help disentangle the value of explicit reasoning (CoT) from the value of the specific instructions within the SysHint prompt.

---

> ### Author Response · Authors · 2025-11-21
> **Rebuttal**
>
> > You claim the two-level (system vs. user) method is "inherently scalable". Could you elaborate on how you envision this working for a more complex, multi-level hierarchy (e.g., System > Tool > User)? Would this require a more complex SysHint prompt, or a dataset with more complex, multi-level conflicts?
>
> Please refer to “General Response, Multi-level Instruction Hierarchies.”
>
> >  The use of RLVR/GRPO is a core part of the methodology. Have you experimented with a simpler SFT approach, where you only fine-tune on high-reward (CoT, Answer) pairs generated from the VerIH dataset? This would help clarify if the full, complex RL loop is necessary or if SFT on curated "good" reasoning traces is sufficient.
>
> For SFT without CoT traces, please refer to “General Response, SFT Results” and “Contributions, Significant Gain.”
>
> As for RLVR vs Self-Distillation with CoT, we do not think it applies to our setup. Self-distillation is better for non-reasoning models. Since our base model Qwen3 already has strong reasoning performance and basic IH reasoning ability with SysHint (refer to Table 1), RL will be a better choice for post-training to enable general IH reasoning. Self-distillation will easily overfit and can not generalize to unseen domains.
>
> >  The VerIH dataset's conflicts are generated by a single LLM (Claude-4-Sonnet). Have you analyzed the diversity of the generated conflicts? Is there a risk that the model has overfitted to the style of conflict that Claude generates, rather than a general concept of instruction conflict?
>
> Please refer to “General Response, Simple Claude-based Rewriting”
>
> >  The w/o CoT_train ablation removes reasoning entirely. What happens in an ablation that keeps CoT generation but removes the specific SysHint prompt? This would help disentangle the value of explicit reasoning (CoT) from the value of the specific instructions within the SysHint prompt.
>
> Please refer to “General Response, SysHint Ablation Study”

---

> ### Author Response · Authors · 2025-11-27
>
> Dear Reviewer i79i,
>
> Thank you for your effort to review our work. We understand that you might be busy recently, but we hope you could take some time to review our rebuttal. We are happy to clarify any points as needed. If our rebuttal addressed your concerns, we would be grateful if you reflected that in an updated score.

---

### Official Review · Reviewer_ZAbp · 2025-10-31

**Soundness:** 2
**Presentation:** 2
**Contribution:** 1
**Rating:** 2
**Confidence:** 4

**Summary:**

The paper introduces a simple data augmentation strategy to improve the instruction hierarchy ability of LLMs. They assume an access to a dataset of system and user prompts such that the responses can be easily verified for alignment. Then, they rewrite the user prompt using a larger LLM to specifically conflict the system prompt. Finally, they train reasoning LLMs with verifiable rewards over the original and conflicting user prompts to enhance their instruction hierarchy and safety.

**Strengths:**

- The paper can be easily followed with coherent writing.
- Experiments show improved performance over the base models and CoT after full fine-tuning.
- Results also show generalization to other benchmarks, particularly to safety-related tasks.
- Ablation of not using the conflicting prompts is also shown.
- Reasoning traces are also qualitatively evaluated to assess the relationship between system and user prompt.

**Weaknesses:**

- Originality is limited since the key benefit and contribution of verifiability of instructions is established from Lambert et al., 2025. On the other hand, the idea of conflicting user prompts is also originally provided in Zhang et al., 2025. Thus, the only contribution is augmenting the RLVR-IFEval dataset with the basic scheme of conflicting user prompts.
- Simple Claude-based rewriting may introduce bias and may not generalize to new types of rewriting structures. More analysis to the diversity of conflicting types should be studied. The examples simply add an additional line which does not seem diverse enough.
- Experiments do not compare with other fine-tuning-based baselines and are limited to prompt-based baselines which is not fair. For example, what is the effect of training on IHEval or the effect of SFT training instead of GRPO?
- Training is limited to reasoning-based LLMs and does not show significant improvement for non-reasoning baselines. More discussion and limitations should be discussed here.
- Verifiable rewards can be problematic due to reward hacking and incorrect reasoning. Reasoning should be qualitatively and quantitatively analyzed as well.
- Ablation on removing SysHint is not provided. Does "+IFEval" include SysHint or not?
- Training is limited to verifiable instruction types (through constraints), and the simple augmentation cannot be easily extended to non-verifiable use cases.
- Minor:
  - Figures 1 and 2 are quite rudimentary and should be updated for better space utilization. Figure 1 should present a true example from the constraint types.
  - Typos in Section 3: RVLR -> RLVR
  - The code is directly copied from verl repository with their metadata (setup.py), which leads a reader to believe that the authors' identities are revealed. This should be updated to avoid any confusion.

**Questions:**

See above weaknesses

---

> ### Author Response · Authors · 2025-11-21
> **Rebuttal**
>
> > Originality is limited since the key benefit and contribution of verifiability of instructions is established from Lambert et al., 2025. On the other hand, the idea of conflicting user prompts is also originally provided in Zhang et al., 2025. Thus, the only contribution is augmenting the RLVR-IFEval dataset with the basic scheme of conflicting user prompts.
>
> The instruction hierarchy (IH) and conflicting user prompts are proposed in Wallace et al., 2024, not Zhang et al., 2025. We are trying to solve the instruction hierarchy problem with reasoning, not propose this concept (line 016 and 089).
>
> As for contributions, please refer to “Contributions”.
>
> > Simple Claude-based rewriting may introduce bias and may not generalize to new types of rewriting structures. More analysis to the diversity of conflicting types should be studied. The examples simply add an additional line which does not seem diverse enough.
>
> Please refer to “General Response, Simple Claude-based Rewriting”
>
> > Experiments do not compare with other fine-tuning-based baselines and are limited to prompt-based baselines, which is not fair. For example, what is the effect of training on IHEval or the effect of SFT training instead of GRPO?
>
> IHEval does not provide a training set. Please refer to “General Response, SFT Results” and “Contributions, Significant Gain.”
>
> > Training is limited to reasoning-based LLMs and does not show significant improvement for non-reasoning baselines. More discussion and limitations should be discussed here.
>
> Please refer to “General Response, SFT Results” and “Contributions, Significant Gain.”
>
> > Verifiable rewards can be problematic due to reward hacking and incorrect reasoning. Reasoning should be qualitatively and quantitatively analyzed as well.
>
> The original manuscript already includes the analysis. Please refer to “Appendix C, IH rate in CoT traces for Qwen3-8B”.
>
> > Ablation of removing SysHint is not provided. Does "+IFEval" include SysHint or not?
>
> Please refer to “General Response, SysHint Ablation Study”.
>
> Yes. “+IFEval” includes SysHint.
>
> > Training is limited to verifiable instruction types (through constraints), and the simple augmentation cannot be easily extended to non-verifiable use cases.
>
> If a simple method with a small dataset can provide a huge improvement, why do you believe a more complex non-verifiable dataset is necessary? Could you provide some examples where such non-verifiable cases offer extra benefits that simple and verifiable data cannot?

---

> ### Author Response · Authors · 2025-11-27
>
> Dear Reviewer ZAbp,
>
> Thank you for your effort to review our work. We understand that you might be busy recently, but we hope you could take some time to review our rebuttal. We are happy to clarify any points as needed. If our rebuttal addressed your concerns, we would be grateful if you reflected that in an updated score.

---

> > ### Comment · Reviewer_ZAbp · 2025-11-28
> >
> > Thank you for your rebuttal! It has helped clarify some of my concerns. However, some major ones remain unresolved, specifically, (1) regarding the originality of the paper that only proposes a simple conflicting user prompt augmentation to the RLVR-IFEval dataset, (2) No further analysis on the conflicting types generated by the Claude model was conducted, (3) No qualitative analysis of reasoning traces was conducted still, (4) Limitation to reasoning LLMs still hold, and (5) it is quite obvious that simple verifiable -> general instruction hierarchy generalization would be harder than generalizing within non-verifiable instruction hierarchy so indeed this training dataset (which is same as RLVR-IFEval) would become a bottleneck at some point (so it's just a weakness not a requirement if that wasn't clear and the authors are recommended to acknowledge this and discuss it effectively). But overall, contributions on top of RLVR-IFEval are very limited in my opinion, and the authors are recommended to expand the contributions by doing a more comprehensive analysis of augmentation strategies of conflicting prompts and simplifying the setup by removing the system prompt (since it doesn't change results in most cases). Furthermore, the SFT comparison is helpful as it shows the benefits of RLVR, but with limited, non-reasoning, and self-generated data. I also don't follow the SFT justification by comparing with another work currently under review in the same conference. Also, the authors have exceeded the paper limit in the revision, which can flag for desk-rejection.

---

> > > ### Author Response · Authors · 2025-12-02
> > >
> > > Thank you for the reply!
> > > However, we believe that our previous rebuttal has addressed the concerns you mentioned here. If there are specific points you disagree with, we would appreciate it if you could identify them directly rather than repeating the original question.
> > >
> > > We do not think it is necessary to remove SysHint. As shown in Table 1. A simple prompt like SysHint can improve performance on IHEval without training. It is natural to include SysHint in RLVR as a fair comparison with the baseline.
> > >
> > > Finally, the rebuttal policy allows up to 10 pages.

---

### Author Response · Authors · 2025-11-21
**General Response**

# Multi-level Instruction Hierarchies
(This part is also updated in our paper, Appendix D: Extending to multi-level instruction hierarchies)

As for evaluating our model on more than two roles, IHEval already has a tool-use subset, which includes three levels: system, user, and tool_calls.

| Tool-Use (Qwen3-8B) | Aligned    | Conflict |
|-|-|-|
| w/o CoT      	| 72.22%  | 4.39%  |
| w/ CoT       	| 84.03%  | 26.60% |
| w/ CoT + SysHint  | 83.04%  | 33.25% |
| +VerIH (Ours) 	| **84.20%**  | **38.66%** |

Although our RLVR training data does not include tool_calls role, the model can generalize to conflict tool-use setup and has a ~5% performance gain.

For further improvement on the multi-level instruction hierarchy setup,  the SysHint can be modified to specify the hierarchy order of instructions:

"Instructions have different priorities: System > User > Tool.

Follow the lower-priority prompt within the bounds of the higher-priority prompt. Think step by step about the relationship among prompts from multiple levels. If there is a conflict, the higher-priority prompt takes precedence."

As for the dataset, there are two ways to extend into multiple levels:
1) Split existing requests/constraints in VerIH into several sub-requests/constraints. Put them at different levels (System, User, Tools, …). Then, randomly select one level and let LLMs rewrite this prompt such that it conflicts with higher-level prompts.
2) Each sample contains only two levels, but different levels (like System vs User, System vs Tool, User vs Tool). After training, we expect the model can generalize from two-level IH reasoning into multi-level IH resolution.

For both approaches, the existing verify functions remain useful. We leave this exploration to future work.

# SysHint Ablation Study
(This part is also updated in our paper, Appendix E: Ablation Study for SysHint)

| Qwen3-8B | IFEval | IFBench | IHEval-Aligned | IHEval-Conflict | MMLU | MATH-500 |
|-|-|-|-|-|-|-|
| + VerIH | 87.41% | 38.21% | 89.89% | 63.48% | 80.63% | 94.20% |
| w/o SysHint_train | 91.85% | 38.51% | 89.41% | 60.73% | 81.58% | 93.60% |

| Qwen3-8B | Harmbench ASR↓ | WildJailbreak Benign↑ | WildJailbreak Harmful↓ | TensorTrust Helpful↑ | TensorTrust Inject↓ |
|-|-|-|-|-|-|
| + VerIH | 1.25% | 97.60% | 41.25% | 86.79% | 32.58% |
| w/o SysHint_train | 2.19% | 97.20% | 42.65% | 89.43% | 52.73% |

SysHint is similar to the CoT prompt “think step by step to solve the question…”, which enables models’ reasoning ability, but it focuses specifically on thinking about instruction hierarchies.

In almost all benchmarks, removing SysHint does not influence the performance.

But for IFEval, w/o SysHint_train improves performance by 4.44%. However, IFBench performance remains the same. One drawback of removing SysHint during training is the increase of ASR in TensorTrust (20.15%), suggesting SysHint enhances generalization to unseen domains during training and helps complex instruction hierarchy resolution. We speculate that future work, which includes safety datasets in IH training, can remedy this issue.

#  SFT Results
Our work proved that even without SFT, the general reasoning ability can be leveraged easily for instruction hierarchy resolution. The reasoning model does not need to be retrained with SFT to get instruction hierarchy reasoning ability (line 21/line 139/line 485). Please refer to “Contributions, Significant Gain” for a comparison of SFT and RL performance.

We also conducted an SFT training experiment as requested by reviewers.
We use Claude to generate the response for the VerIH dataset, resulting in ~5K question-answer pairs.
All answers are filtered by verifiable functions to ensure the quality of the training set. Out of ~7K questions, Claude can only generate a correct response for ~5K questions in 5 attempts.
The training schema is the same as RLVR. And the results are shown here:
| Qwen3-8B | IFEval | IFBench | IHEval-Aligned | IHEval-Conflict | MMLU | MATH-500 |
|-|-|-|-|-|-|-|
| + VerIH | 87.41% | 38.21% | 89.89% | 63.48% | 80.63% | 94.20% |
| +SFT | 63.55% | 11.64% | 49.56% | 30.97% | 57.38% | 47.20% |

| Qwen3-8B | Harmbench ASR↓ | WildJailbreak Benign↑ | WildJailbreak Harmful↓ | TensorTrust Helpful↑ | TensorTrust Inject↓ |
|-|-|-|-|-|-|
| + VerIH | 1.25% | 97.60% | 41.25% | 86.79% | 32.58% |
| +SFT | 16.88% | 86.80% | 68.75% | 43.96% | 56.18% |

SFT training does not help at all for model performance in IH and safety-related tasks, but causes a significant drop in general tasks like MMLU and MATH-500.

#  Simple Claude-based Rewriting
We argue that the simplicity of our dataset is a merit, not a flaw. Even with such simplistic Claude-based rewriting, we obtain huge improvements on IH benchmarks in addition to generalizing to the safety domain, data for which was not included during training (line 025). A more complex/diverse dataset might further improve the performance and is a good avenue for future work.

---

### Author Response · Authors · 2025-11-21
**Contributions**

1. **Reframe IH.** We propose reframing instruction hierarchy as an explicit reasoning problem (line 089). Instead of directly learning input-response mapping, the model reasons about the relationship between higher-level (system) and lower-level (user) prompts before generating a response.

2. **Simple Training.** While other papers (Wallace et al., 2024; Wu et al., 2024b) use SFT for IH, we use RLVR instead. SFT requires large amounts of input-response data (221k on HieraSuite https://openreview.net/pdf?id=gMajoi2xsq; >210K on Wu et al., 2024b). While generating responses for SFT is costly, producing reliable reasoning traces is even more challenging. Another limitation is that conducting SFT on an already instruction-tuned model is susceptible to forgetting, leading to a decline in general model capabilities (https://arxiv.org/abs/2510.18874, https://arxiv.org/abs/2404.18466, https://arxiv.org/abs/2511.08567).

    We only use ~7K IFEval-style training data with verifiable answers. Our method does not need reasoning traces or responses as supervision (line 20). It is, in fact, simpler than SFT.

3. **Significant Gain.** After RLVR training on VerIH dataset, our model has ~20% performance gain on IHEval conflict setup (Table 1). As a comparison, we find a contemporary paper HieraSuite (https://openreview.net/pdf?id=gMajoi2xsq), which conducts SFT with 221k examples. On the IHEval benchmark, our Qwen3-4B model (+VerIH)(aligned 87.04%, conflict 57.21%) has better performance than their Qwen2.5-14B-IT (+HieraCRO) model (aligned 83.7%, conflict 52.5%). Qwen3-8B model (+VerIH)(aligned 89.89%, conflict 63.48%) has similar performance to their Qwen2.5-32B-IT (+HieraCRO) model (aligned 88.0%, conflict 65.2%). We leverage the existing reasoning ability of the model for IH, without losing general reasoning performance.

4. **Generalization.** Previous work (Wallace et al., 2024; Wu et al., 2024b) focuses on the safety domain by specifically creating training and evaluation datasets on safety. We believe IH is a more general question. The user prompt can be in conflict with the system prompt, but also be harmless. And security can be viewed as a special case of the conflict setup in IH  (line 236).

    Instead of training on safety-related datasets like their work, our VerIH dataset is about verifiable constraints on response format, quantity, and keyword usage (line 098). Although no safety-related data is included during RLVR, our method can generalize to safety-critical tasks, providing up to 20% improvement on prompt injection and jailbreak benchmarks, without losing general performance on begin request (Table 2).

---

### Note · Authors · 2026-01-05

**Comment:**

We would like to withdraw our submission with the agreement of all authors.
Thank you for your time.

**Withdrawal Confirmation:**

I have read and agree with the venue's withdrawal policy on behalf of myself and my co-authors.